# Inertial Tracking System for Monitoring Dual Mobility Hip Implants In Vitro

**DOI:** 10.3390/s23020904

**Published:** 2023-01-12

**Authors:** Matthew Peter Shuttleworth, Oliver Vickers, Mackenzie Smeeton, Tim Board, Graham Isaac, Peter Culmer, Sophie Williams, Robert William Kay

**Affiliations:** 1Future Manufacturing Processes Research Group, University of Leeds, Leeds LS2 9JT, UK; 2Institute of Medical and Biological Sciences, University of Leeds, Leeds LS2 9JT, UK; 3Wrightington, Wigan & Leigh NHS Foundation Trust, Wigan WN6 9EP, UK; 4Healthcare Mechatronics, University of Leeds, Leeds LS2 9JT, UK

**Keywords:** implants, inertial tracking, sensor fusion, orientation measurement, hip replacement, dual mobility

## Abstract

Dual mobility (DM) implants are being increasingly used for total hip arthroplasties due to the additional range of motion and joint stability they afford over more traditional implant types. Currently, there are no reported methods for monitoring their motions under realistic operating conditions while in vitro and, therefore, it is challenging to predict how they will function under clinically relevant conditions and what failure modes may exist. This study reports the development, calibration, and validation of a novel inertial tracking system that directly mounts to the mobile liner of DM implants. The tracker was custom built and based on a miniaturized, off-the-shelf inertial measurement unit (IMU) and employed a gradient-decent sensor fusion algorithm for amalgamating nine degree-of-freedom IMU readings into three-axis orientation estimates. Additionally, a novel approach to magnetic interference mitigation using a fixed solenoid and magnetic field simulation was evaluated. The system produced orientation measurements to within 1.0° of the true value under ideal conditions and 3.9° with a negligible drift while in vitro, submerged in lubricant, and without a line of sight.

## 1. Introduction

Dual mobility (DM) hip replacements differ from conventional hip replacements in that there are two mobile bearing surfaces as opposed to one. The design consists of a spherical femoral head that can articulate in a hemi-spherical polyethylene (PE) liner that can in turn articulate within a metallic acetabular shell (Figure 1a). Motion is predicted to occur predominantly at the femoral head due to the lower frictional torque produced by its smaller surface area, engaging the outer, acetabular shell bearing when the stem’s neck impinges on the PE liner at extremes of motion (Figure 1b–d).

These characteristics offer an enhanced range of motion and a greater stability compared with conventional implants and, therefore, are often used for patients at a high risk of dislocation, such as those with neuromuscular conditions, abductor deficiencies, or for THR following a tumor resection [1]. DM bearings have been used increasingly in recent years. According to the American Joint Replacement Registry [2], in 2020, DMs were used in approximately 10% and 20% of all primary and revision total hip replacements (THR) performed in the USA, respectively. As their use continues to increase and indications widen, understanding the mechanics of these implants will become more important and could offer an insight into the long-term function and potential failure modes of DMs. This in turn could lead to enhancements in the implant design, material selection, surgical technique, and indications for use, thus improving the surgical outcomes.

To achieve the suitable monitoring of liner motions while undergoing in vitro testing, a tracking system would be required to operate while the joint is submerged in a lubricant, impeding any line-of-sight measurement methods. Furthermore, unlike conventional hip replacements, the motions of joint simulators cannot be used to indicate a liner orientation, as the liner is free-to-rotate. Previous work has explored the addition of telemetry to conventional THR implants to remotely observe the temperature and forces [3,4,5] acting upon the femoral components; however, while some of these approaches have been used in patients and are, therefore, capable of operating in the required conditions, none have been used in DM implants or to provide an orientation measurement [6].

In its simplest form, inertial orientation tracking uses three-axis accelerometers to provide two degree-of-freedom orientations measurements relative to gravity without the use of a visible reference. While suitable for a static angular measurement, this approach is sensitive to external acceleration from vibrations, knocks, or translations which become superimposed on the gravitational readings. To overcome this, sensor fusion algorithms take additional data from gyroscopes and magnetometers, using them to converge on a more reliable orientation estimate and reducing the effects of external disturbances [7]. In 2008, Mahony et al. [8] reported a fusion algorithm tailored towards low-cost inertial measurement units (IMU) that could produce stable attitude references from noisy accelerometer and gyroscope data. This was improved upon by Madgwick et al. [9] who included magnetometer data to provide full attitude and heading references. Additionally, the algorithm removed the implementation complexities associated with traditional Kalman filters, such as the need for an accurate model of measurement errors in the system, while maintaining a comparable degree of performance and a high degree of computational efficiency.

In recent years, inertial tracking has been applied to consumer electronics such as cell phones [10,11] and unmanned arial vehicles [12,13,14], which has driven manufacturers to develop highly miniaturized microelectromechanical system (MEMS)-based IMUs, containing accelerometers, gyroscopes, and magnetometers [15]. Current generation MEMS IMUs are now available in integrated-circuit (IC) package footprints as small as 3 mm × 3 mm and can provide full nine degree-of-freedom measurements.

IMUs have already been used in the field of orthopedics and biomechanics for applications including gait analysis [16], surgical tracking [17], and a component implantation angle [18]. This study aimed to develop and validate an IMU-based system for tracking the orientation of dual mobility implant liners relative to the acetabular shell while the implant was inside a six degree-of-freedom hip joint simulator submerged in lubricant. The contributions of the work were the design of a bespoke IMU-based tracker circuit board, the use of a solenoid coupled with magnetic simulation to enable field tracking in magnetically noisy environments, and the validation of the above under both idealistic and realistic test conditions. Additionally, the article details a method for performing a calibration for all on-board sensor packages to improve the accuracy of the trackers.

## 2. System Design

### 2.1. Trackers

Before designing the tracking system, a set of key requirements were defined to ensure the system could operate in vitro within a typical hip simulator environment. The system should:Operate without requiring a line of sight;Be able to operate immersed in lubrication fluids;Mount to a DM implant without affecting the bearing surfaces and overall implant motion;Have a stable operation without excessive sensor drift to allow for the observation of trends in motion;Be resistant to magnetic disturbances.

For this study, excessive sensor drift was defined as a continuous directional shift in the tracker’s bias of more than 0.05 °/s over the course of a test. The type of hip simulator used for the final validation (Anatomical Hip Simulator, Simulation Solutions, Unit 10, Rugby Park, Bletchley Road, Stockport, UK) had all three rotational degrees of freedom on the femoral side of the implant and, therefore, the cancellation of hip rotation is not required. The selected IMU (ICM-20984, TDK InvenSense, 1745 Technology Drive, San Jose, CA, USA) was chosen as it had the smallest available footprint, using a quad-flat no-leads-24 (QFN-24) IC package, while also providing performance characteristics suited to the low frequency, which is typically less than 1.5 Hz (Table 1).

The tracking system consisted of three modules, one on the implant; one on the tracker and two outside of the test environment; and one on the controller and the host. The only suitable space for the tracker on a PE liner that that would not impede the DM implant function was a flat, annular face outside the two bearing surfaces as shown in Figure 2. To fit this area, a custom, C-shaped printed circuit board (PCB) was designed with an inner radius of 37.0 mm and an outer radius of 52.5 mm. This allowed the tracker to mount on PE liners with a bearing outer diameter of 69 mm and a femoral head size of 28 mm or smaller. In addition to the IMU, the PCB carried voltage regulation and logic level translation circuits for conversion between 1.8 V for the IMU and the 3.3–5 V used by the microcontroller unit (MCU). Communication and power to the tracker was provided by a 750 mm long, 4-wire I2C tether directly soldered to the PCB. To allow the tracker to be submerged, the PCB was coated in a water-resistant epoxy. The final tracker design was bonded to the PE liners using double-sided adhesive tape (668 Double Coated, 3M).

For the controller module, A 600 MHz, 32-bit MCU (Teensy 4.0, PJRC) was used to communicate with the IMU, perform general I/O, and to implement the Madgwick fusion algorithm [9] and calibration calculations. Once calculated, the MCU transmitted timestamped orientations in the Tait–Bryan convention to a host PC via a USB link, where it was logged by a custom-written interface and data logging program.

### 2.2. Calibration

MEMS IMUs offer significant size and power savings over mechanical and optical variants at the expense of measurement accuracy [20]; however, this can be improved by calibrating the sensor packages within the IMU on a chip-by-chip basis. This was achieved by subjecting the trackers to a series of controlled static and dynamic tests while measuring the raw sensor outputs to calculate the calibration factors. These were applied to the data prior to the sensor fusion. A desktop-mounted robot arm (UR3, Universal Robots GmbH) was used to manipulate the trackers while gathering the calibration data. For the alignment of the reference frames, the trackers were affixed to a jig (Figure 3) that could be rotated about the tool flange’s Z-Axis and locked in place.

A software offset was applied to the tool flange reference frame to ensure that the robot performed a motion planning calculation about the trackers center of rotation. The automation of the calibration using a robot arm provided a greater repeatability than could be achieved by manual manipulation and allowed for the calibration of all sensors without transitioning between different equipment. The setup offered a high degree of flexibility allowing for unique calibration motions for the accelerometers, gyroscopes, and magnetometers, as well as the final test profiles to be performed without hardware changes.

The individual sensor packages were calibrated in the following sequence:Accelerometers;Gyroscopes;Magnetometers.

This order allowed the accelerometers to be used for an alignment relative to gravity for subsequent calibration stages. After each stage, new calibration values were stored in the MCU.

#### 2.2.1. Accelerometers

A commonly used accelerometer calibration procedure requires the IMU to be held static in six positions with a precise colinear alignment of each axis to the gravity vector [21], however, without high-quality positioning equipment, this can be challenging. To overcome this, a Gauss–Newton nonlinear optimization approach [22] was applied to 3D acceleration data gathered while holding the tracker static in twelve unique orientations. For each orientation, 5 s of data was averaged to reduce the effects of the signal noise. From this, the algorithm produced a 3 × 1 bias correction vector and a 3 × 3 scale and cross-sensitivity matrix.

#### 2.2.2. Gyroscopes

To minimize the off-axis effects of MEMS gyroscope g-sensitivity [23], the tracker was aligned such that its Z-axis was aligned with gravity. This was achieved by monitoring the calibrated accelerometer outputs and adjusting the ϕ and θ angles until the X and Y axes showed a 0 g response. Subsequently, a 90° ϕ rotation was performed, followed by a ψ rotation, until the Y and Z axes showed the same 0 g. With all axes aligned about the gravitational vector, a −90° ϕ rotation was applied so that X and Y read 0 g and Z read −1 g.

Once aligned, the tracker was held static for 30 s and oscillated about each axis individually while recording the angular rates. The rotations are detailed in Figure 4.

The gyroscope bias for each axis was defined as the mean angular rate during the static portion of the routine. These were then subtracted from all the raw data before integrating the dynamic regions, with respect to the time, to find the angle rotated as measured by the gyroscopes. Finally, the scale factors were calculated by dividing the actual angle by the measured angle for each axis.

#### 2.2.3. Magnetometers

For an ideal magnetometer in a uniform magnetic field, plotting the X, Y, and Z magnetic field strengths for a range of orientations should produce a sphere of points centered on the origin with a radius equal to the field strength. Errors in the magnetometer and distortions to the local magnetic field transform the sphere. This distortion was measured and compensated for using the magcal function provided by MATLAB’s Sensor Fusion and Tracking Toolbox [24].

To collect magnetic field data, the robot arm rotated the trackers 5° in ϕ, followed by 360° in ψ. This pattern was repeated until the board was inverted. The direction of the ψ rotations alternated to prevent the tether becoming tangled.

The algorithm produced a 3 × 1 bias correction vector and a 3 × 3 scale and cross-sensitivity matrix. This calibration method could also be performed manually in the test-environment without the aid of a robot arm to account for localized magnetic distortions, provided they were constant in nature.

### 2.3. Magnetic Distortion Tolerance

Hip joint simulators are used for the preclinical testing of hip replacements and are designed to replicate walking by applying dynamic motion and loading cycles to an implant. During initial testing, a strong magnetic interference was observed when the trackers were placed in some hip simulators, leading to a random drift about the *Z*-axis when static. Through a measurement, this was attributed to large quantities of ferrous materials and stray magnetic fields from nearby power electronics. Additionally, these disturbances fluctuated and were non-uniform, meaning they could not be accounted for with a simplistic, static calibration model.

To solve this, a magnetic solenoid (Figure 5) that remained stationary during the testing was introduced to generate a stable reference for the magnetometers. The solenoid consisted of 125 turns of a 0.51 mm diameter-enameled copper wire wrapped around a 79 × 16 mm (diameter × height) bobbin which was 3D printed from polyethylene terephthalate glycol (PETG). During hip simulator testing, the solenoid was mounted to existing bolt holes in the acetabular shell fixture.

The solenoid produced a toroidal-shaped magnetic field with a strength that varied depending on the position within the field. Since the fusion algorithm requires a uniform field strength, a linearization technique was employed in which the solenoid was computationally simulated in Python to produce a lookup table of the expected field strength components and their equivalent values in a uniform field. This table was stored on the MCU which used a ‘nearest neighbor’ search to substitute the measured values with the equivalent uniform field strengths prior to the sensor fusion.

## 3. Experimental Validation

The analysis of the trackers was split into two studies, an initial evaluation under ideal conditions to produce a baseline, followed by in vitro testing with lubricant and magnetic interference to explore the impact of using the trackers in their design environment.

### 3.1. Tracker Performance under Ideal Conditions

#### 3.1.1. Method and Materials

The tracker’s performance was validated by moving the calibrated units through a series of motions that excited all three IMU axes simultaneously and by comparing the tracker and robot arm orientations.

The testing was performed on the robot arm setup detailed in Section 2.2, and a real-time data exchange (RTDE) was used to timestamped tool flange orientations at 125 Hz. A modified python script from the manufacturer [25] was used to trigger the RTDE and log the results to a file on the trackers’ host PC. Each test consisted of 60 cycles whereby the robot sequenced through four different poses. These were selected such that each axis oscillated about the origin by a specified range of motion (ROM). The speed of the axes was dictated by setting the period for the cycle, and the motion planning between each pose was automatically managed by the robot’s inverse kinematics. An example of how test motions were timed is shown in Figure 6.

To explore the individual effects of the cycle period (t) and ROM on the positional accuracy, each was varied independently. When evaluating the effect of an increasing cycle period, the ROM was fixed at ±20° as this was slightly greater than the angles a DM implant’s liner would be expected to experience in vitro. The robot’s motion control software allowed for minimum cycle period increments of 0.4 s, with the shortest achievable period being 1.2 s for a ROM of 20°. A maximum value of 2.8 s was chosen, to give five different conditions.

When testing variations in ROM, the average angular speed ($\omega_{AVE}$) was kept constant to remove the secondary influence from potential inconsistencies in the gyroscopes angular rate measurements and was defined as the total 3D angle swept through per cycle divided by the cycle period. The average angular speed could not be directly set by the robots motion control software; therefore, it was controlled indirectly using the relationship between the average angular speed, ROM, and cycle period. Using these values and the definition of the average angular speed, the 3D angle that would be swept through in each test could be found using Equation (1).
(1)$$\theta =t\cdot \omega_{AVE}$$
where:
θ was the swept angle in °t was the period for one cycle in s$\omega_{AVE}$ was the average angular speed in °·s^−1^

Because the rotations occur in 3D, the swept angle and the ROM for each axis are not equal, however, by converting the four poses to quaternions, the angular distances between each can be found and summed up to find θ or a particular ROM Equation (2).
(2)θ=∑i=1n2·cos−1ℝpi·qi¯·180π
where:

n was the number of segments in the cycle

$p_{i}$ was the segment start position in quaternions

$q_{i}$ was the segment end position in quaternions

By calculating this for multiple ROM values, the relationship between the ROM and θ was shown to be linear and equal to:(3)θ=4.82·ROM+0.03

Finally, combining Equations (1) and (3) shows that the ROM required to maintain a constant average angular speed for a particular cycle period could be represented using Equation (4):(4)$$ROM=\frac{\left(t\cdot \omega_{AVE}\right)-0.03}{4.82}$$

It is important to note that this relationship was only true for the motions used in these tests and would have to be recalculated should the sequence be adjusted.

For evaluating a fixed $\omega_{AVE}$, 41.5 °·s^−1^ was chosen and the same t values from the constant ROM testing were used as they were known to run within the performance limits of both the robot arm and the hip simulator. The known robot poses were defined in the Tait–Bryan angles from the ROM and the sequence shown in Figure 6 and were then converted to quaternions using the eul2quat function of the MATLAB Robotics Systems toolbox [26]. The final test parameters are detailed in Table 2.

MEMS IMUs can be susceptible to manufacture-induced measurement errors from stresses imparted to the die during the soldering process [27,28]. Theoretically, the calibration routine would account for this error, however, to observe the impact of tracker-to-tracker variation, the tests were repeated across five individual tracker PCBs of the same design. A video overview of the experimental setups from this study can be found in the Appendix A.

#### 3.1.2. Results

To evaluate the angular accuracy of the trackers, the maximum and minimum peak angles from all three axes were extracted from the datasets automatically based on their prominence. The resulting values were used to calculate the absolute error between the tracker and robot arm for each axis. The value from the individual trackers were then collated to produce Figure 7, which shows the variation in the positional error for each rotational axis with increasing time periods and ranges of motion. The error bars represent the standard deviation of the peak error values. A positive value indicated that the tracker measurements had overshot the true value.

Under ideal conditions, a similar performance was observed for Φ and θ rotations, with the mismatch between the robot and the tracker reaching a worst-case average of 0.2°. Across all trackers, the standard deviation for these rotations remained below 0.6° with no measurable trends with respect to the angular speed or the range of motion. Similarly, there were no trends observed for the ψ rotations, however, the peak measurement error was greater with a worst-case mean of 1.0°. This error was consistently positive, indicating that rotations about the *Z*-axis typically overestimated the true value. The precision was also reduced with a maximum standard deviation of 1.9°. This behavior was expected as the *Z*-axis is colinear with the gravitational vector, therefore, any ψ rotations will create no change in the accelerometer readings. This results in one fewer reference for the Madgwick algorithm, meaning that the ψ rotations are calculated purely from the nosier magnetometer readings and are more drift-prone gyroscope measurements. All data produced during this test and the associated MATLAB scripts can be found in the University of Leeds Data Repository (https://doi.org/10.5518/1232).

### 3.2. Tracker Performance In Vitro with Lubrication and Magnetic Interference

#### 3.2.1. Method and Materials

To verify that the tracker could perform under the design conditions, the same five trackers were re-calibrated and loaded into a 6-DOF hip simulator (Anatomical Hip Simulator, Simulation Solutions, Unit 10, Rugby Park, Bletchley Road, Stockport, UK) (Figure 8) for testing while submerged in lubricant, without a line of sight. To ensure that the simulator and tracker moved as one, a locking ring that pressed onto the flat surface of the liner was used to disable the femoral head articulation, allowing only the acetabular articulating surface to rotate. To demonstrate performance changes due to a magnetic interference, the tests were repeated with and without the solenoid installed. The solenoid was powered from an external power supply at a constant 1.6 V, consuming 0.45 A and producing a theoretical field strength of 6.1 mT at its center.

The acetabular shell was cemented into a fixture that remained static, while the femoral head and PE liner were mounted to a spigot attached to the simulator’s mobile rotational axes. The tracker was adhered to the liner using double-sided adhesive tape and the tether threaded through access ports in the acetabular fixture.

To align the simulator and tracker coordinate systems, the simulator was sent to an abduction angle of 20°. Then, while the tracker was rotated about its *Z*-axis, the calibrated acceleration values were observed until the X acceleration read zero, signifying that the *X*-axis was perpendicular to gravity and aligned with the flexion axis of the simulator.

Finally, a flexible gaiter was sealed around the assembly and filled with phosphate-buffered silane (PBS) to act as a lubricant. The PBS salt content damages the electronic components that are not protected from liquid ingress, so this test also gives an indication of whether the encapsulation process was successful.

A triaxial motion was achieved by applying a sinusoidal waveform to each axis and temporally shifting them so that each was 45° out of phase with each other. The tests consisted of 60 cycles and varied the average angular speed and range of motion to separate their individual effects. Because the speed was dictated by the cycle time period and swept angle, the method detailed in Section 3.1 was used to find values that kept the average angular speed constant where necessary. Due to the range of motion and velocity constraints of the simulator and the robot arm, the same conditions could not be used on both; instead, a test envelope that produced the least difference between setups was used. The final test variables are detailed in Table 3. A video overview of the experimental setups from this study can be found in the Appendix A.

#### 3.2.2. Results

The approach described in Section 3.1.2 was applied to the output from the trackers and feedback from the simulator to produce Figure 9 and Figure 10. The first shows the results of the tests where the solenoid was not used, leaving the trackers susceptible to an external magnetic interference. The second shows the same tests but with the solenoid energized and the MCU configured to rely on the internal lookup-table generated by the magnetic field simulation.

Without the solenoid (Figure 9), all axes of rotation exhibited an increase in spread when compared to ideal conditions, with each showing differing levels of precision. Φ, θ, and ψ rotations had maximum standard deviations of 1.5°, 1.2°, and 4.6°, respectively. Despite this, the mean error remained consistent and low for φ, θ reaching a maximum of 0.2° and <0.1°, respectively. Ψ rotations were more erratic, with the mean reaching 4.2°.

Following the tests, the readings of the local magnetic field inside the working space of the hip simulator were taken which revealed that the distortion of the magnetic field was such that the magnetic field vector was orientated almost parallel to the gravity vector.

This meant that the fusion algorithm had zero absolute positional references for rotations about the *Z*-axis and, as a result, would have been based on the integration of the angular rates. These rates were provided by gyroscopes, which are known to drift over time, leading to the increased spread in the data. The measurements of the local magnetic field while the simulator was moving were not possible, but the electromagnetic noise it produced may also have contributed to the spread.

Using the solenoid (Figure 10) lead to noticeable improvements in the precision across all axes, bringing the maximum standard deviation for the φ, θ, and ψ rotations down to 0.7°, 0.5°, and 1.0°, respectively. These results were in line with the findings of the ideal conditions study. Once again, the average error for φ and θ were similar, with a maximum of 0.4°, with little variation due to the angular speed or range of motion.

With the solenoid, the ψ rotations appeared to demonstrate a ROM-dependent overshoot reaching a maximum of 3.9° for a ROM of ± 20°. This was apparent from the increasing trend in the ROM tests and the constant error in the angular speed tests which used a constant ROM. Furthermore, the error during the angular speed tests fell on the trendline of error vs. ROM. The likely cause of this was differences between the magnetic simulation and the physical test setup, either due to manufacturing tolerances or the accuracy of the placement of the tracker on the PE liner. It should be possible to introduce a post-mounting calibration routine that adjusts the lookup table based on a set of readings taken before testing. The aim of this would be to account for any setup inaccuracies.

By adding the positive and negative peak measurements from a single cycle, the bias for that cycle was evaluated and then used to evaluate the change in bias over the course of a test. The results for all the tests are shown in Figure 11, separated by their use of the solenoid. Only values for ψ rotations are shown as no drift was observed on the other two axes.

Noticeably, more erratic behavior was observed when the solenoid was not used with a maximum change in the ψ bias of 8.3° over 60 cycles, indicating a significant drift in the zero value from the start of the test. Additionally, the magnitude and direction of the drift was inconsistent and therefore would be difficult to account for with generalized mathematical models. Conversely, introducing the solenoid reduced the average change in the ψ bias to near zero for all tests, with any measurable variation being attributable to the noise in the magnetometer readings. All data produced during this test and the associated MATLAB scripts can be found in the University of Leeds Data Repository (https://doi.org/10.5518/1232).

## 4. Conclusions

This study has successfully demonstrated, for the first time, the application of MEMS IMUs and sensor fusion for monitoring dual mobility hip implant motions, without a line of sight and under laboratory-simulated, typical operating conditions. This is important as it provides a method for a better determination of the function and potential failure mechanisms of an implant that is being used in increasing numbers and seeing widening indications for use. The technology detailed in this work will enable clinically relevant studies to be performed on DM implants, hopefully leading to improved implant designs, updated surgical techniques, and better indications for when DM implants should be used.

Additionally, this technology provides a preliminary understanding for smart, telemeterized implants that could provide patient feedback about the performance and success of their total hip replacement; although, in its current form, the system has two notable limitations that would need to be addressed before it could be considered for use in vivo. The first is the use of a tether for power and communications, as passing wires out of a patient is an invasive and unrealistic solution. Second, the measurements were relative to a static reference frame, meaning that if the acetabular shell moves from its initial orientation, for example due to a hip rotation when waking, the true rotations of the liner are not known. The introduction of an acetabular tracker could be used to mitigate these effects.

In the future, further studies into the long-term performance of the trackers, and the mechanisms behind the ROM-dependent error when using the solenoid, will help improve the performance.

## Figures and Tables

**Figure 1 sensors-23-00904-f001:**
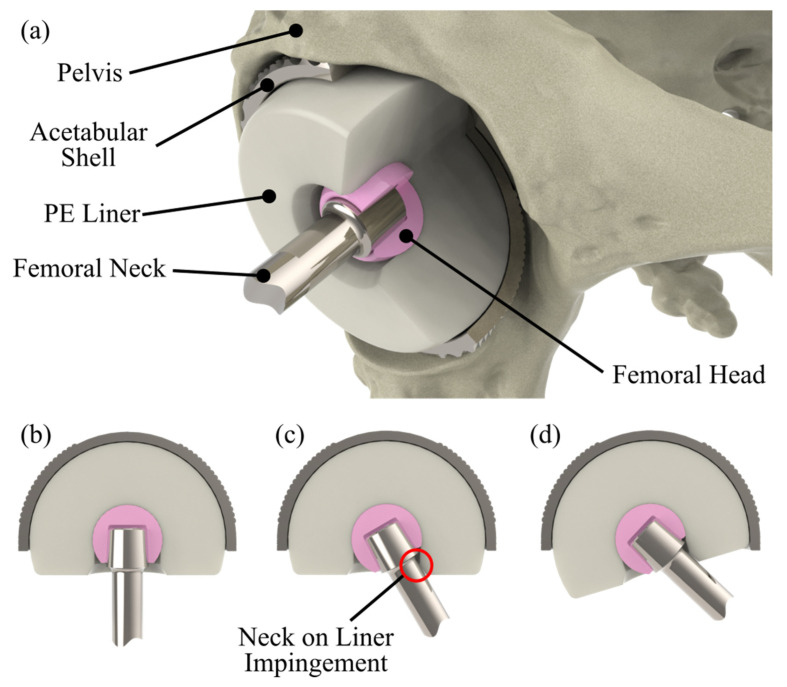
Cutaway images of a dual mobility implant illustrating (**a**) the implants construction, (**b**) its unarticulated position, (**c**) articulation at the inner bearing surface highlighting the location of neck-on-liner impingement which, with further movement, leads to (**d**) articulation at the outer bearing.

**Figure 2 sensors-23-00904-f002:**
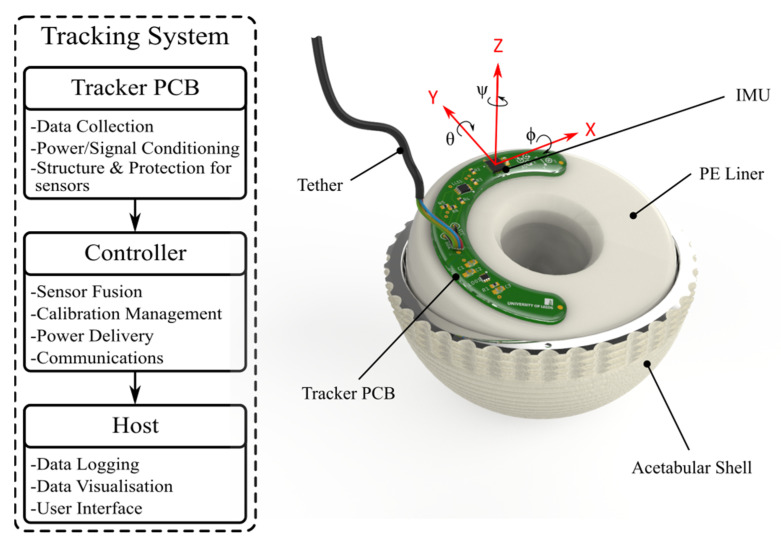
System level flow diagram showing the three main modules of the tracking system concept, and a PCB level render illustrating the main components of the tracker and showing how the PCB mounts to a DM liner.

**Figure 3 sensors-23-00904-f003:**
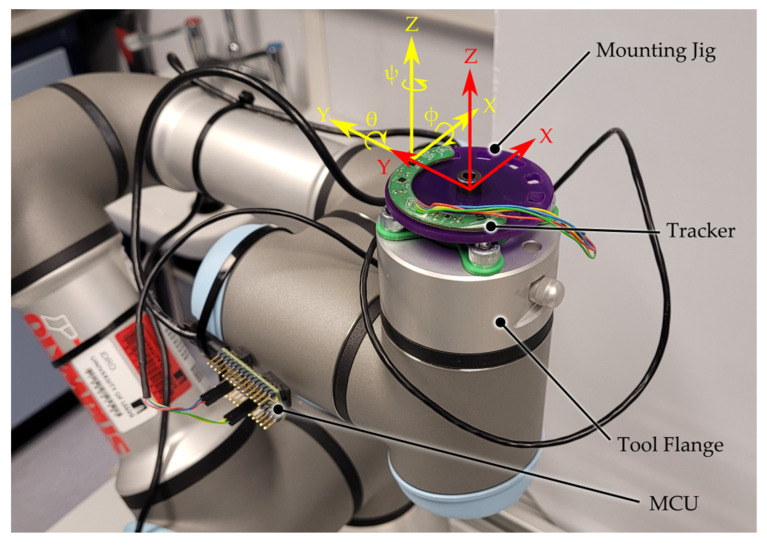
Labelled photograph showing the experimental setup of the tracking system on the tool flange of the robot arm. The yellow axes denote the IMU reference frame which, once fully calibrated, was aligned with the tool reference frame (red).

**Figure 4 sensors-23-00904-f004:**
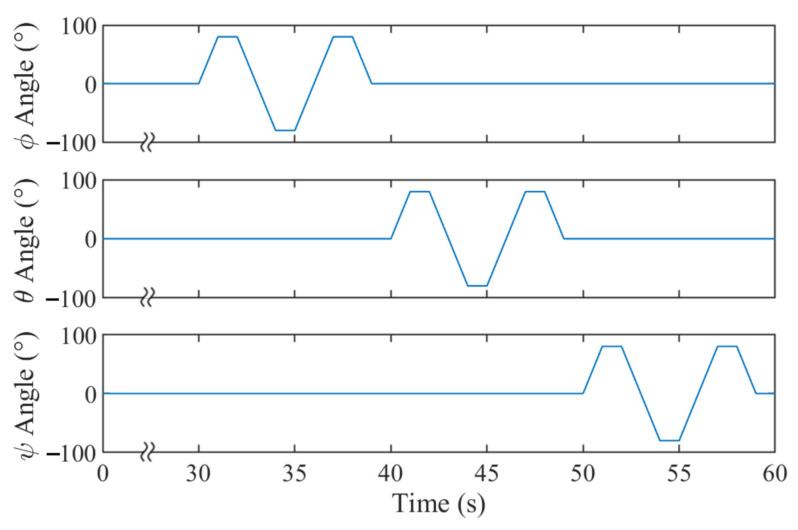
Plots showing the rotations about all three axes that were applied to the trackers arm for calibrating gyroscopes. The motions were performed by the robot arm in the trackers reference frame.

**Figure 5 sensors-23-00904-f005:**
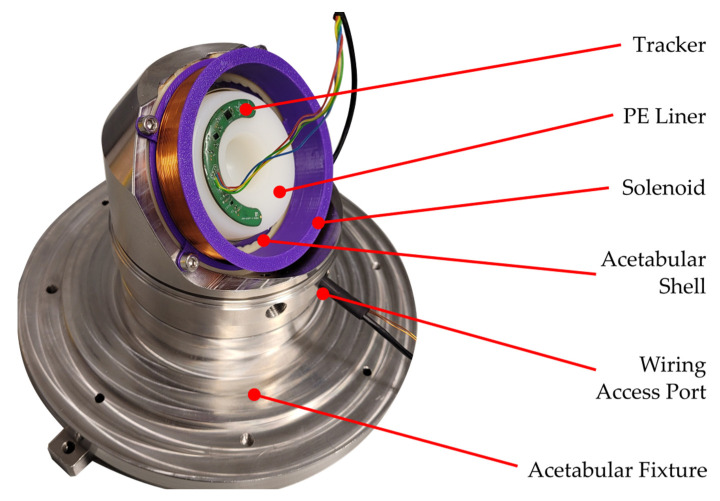
Photograph of the tracker and solenoid mounted to a DM implant cemented into the acetabular fixture used for in vitro testing with the key features labelled. The tether from the tracker passes between the solenoid and the wall of the fixture and out through the access port along with the power for the solenoid.

**Figure 6 sensors-23-00904-f006:**
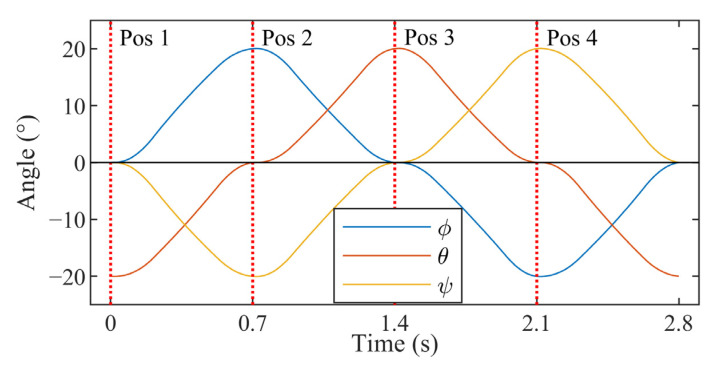
Plot showing the robot arm orientations throughout one example cycle. The red dotted lines represent the times at which the robot was at one of the orientations defined by the test plan.

**Figure 7 sensors-23-00904-f007:**
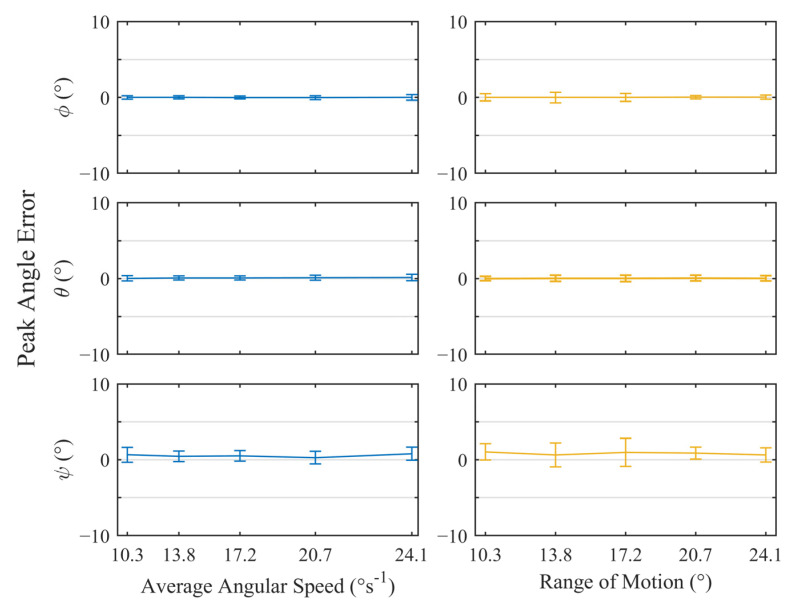
Plots of the mean angular error ± standard deviation for all peaks across n = 5 trackers, while undergoing test motions under ideal conditions. The left- and right-hand plots illustrate variation of error with respect to average angular speed and range of motion, respectively.

**Figure 8 sensors-23-00904-f008:**
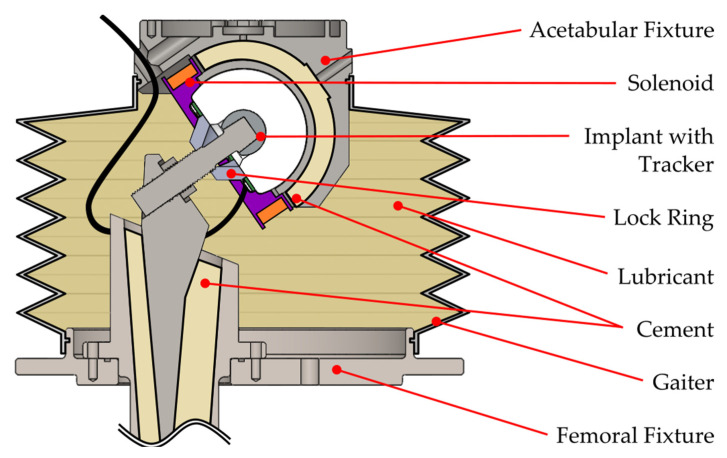
Schematic of the fixture setup used for validating the tracker performance in vitro. See Figure 5 for further solenoid mounting details.

**Figure 9 sensors-23-00904-f009:**
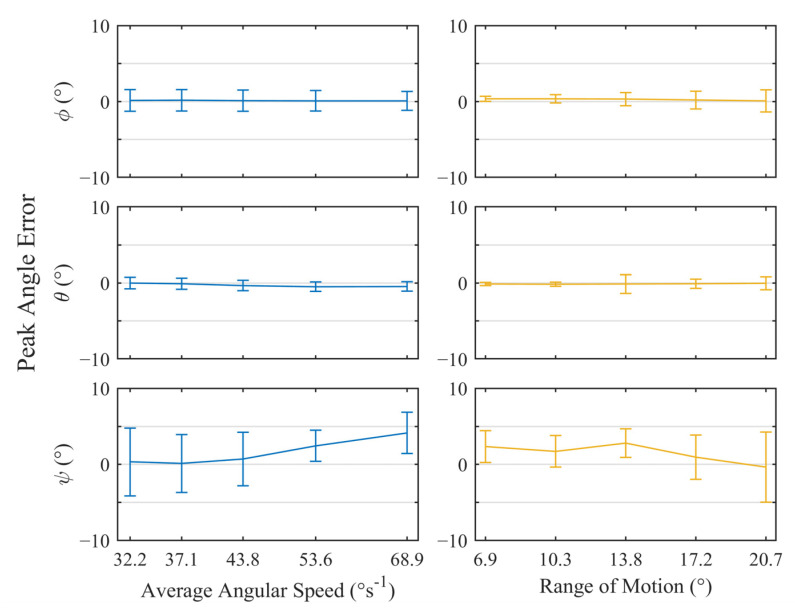
Plots of the mean angular error ± standard deviation for all peaks across *n* = 5 trackers while submerged in lubricant without the assistance of the solenoid. The left- and right-hand plots illustrate variation of error with respect to average angular speed and range of motion, respectively.

**Figure 10 sensors-23-00904-f010:**
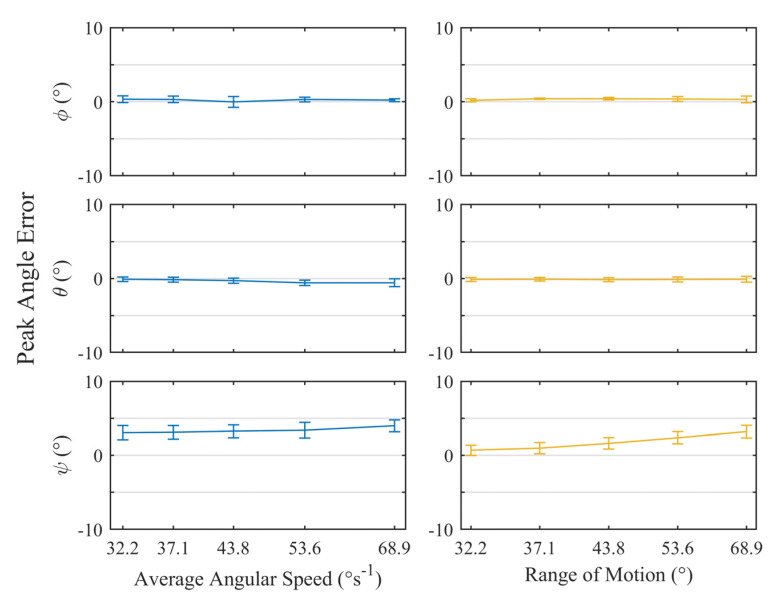
Plots of the mean angular error ± standard deviation for all peaks across *n* = 5 trackers while submerged in lubricant with the addition of the solenoid. The left- and right-hand plots illustrate variation of error with respect to average angular sped and range of motion, respectively.

**Figure 11 sensors-23-00904-f011:**
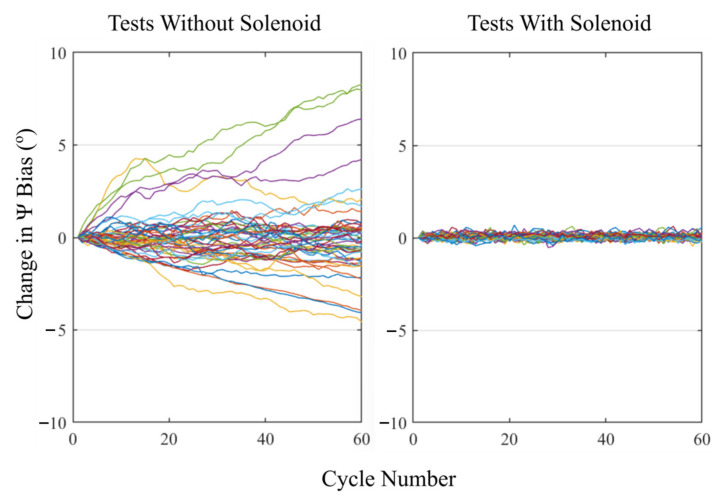
Plots showing the change in bias for rotations about the *Z*-axis as the tests progressed, for all tests. The left-hand plot illustrates the results for tests without the solenoid and the right-hand plot illustrates the results with the solenoid. Each line represents one test on one tracker, resulting in 50 lines per plot.

**Table 1 sensors-23-00904-t001:** TDK InvenSense ICM-20948 Technical Specifications [19].

Characteristic	Accelerometers	Gyroscopes	Magnetometers
Full-Scale Range	±2, 4, 8, 16 g	±250, 500, 1000, 2000 dps	±4900 µT
Resolution	61, 122, 244, 288 µg	0.008, 0.015, 0.030, 0.061 dps	0.075 µT
Data Output Rate (Hz)	4.5 k	9 k	100

**Table 2 sensors-23-00904-t002:** Robot Arm Test Parameters.

Test Variable	Range of Motion (±°)	Average Angular Speed (°/s)	Cycle Period (s)	Angle Swept per Cycle (°)
Average Angular Speed	20.0	80.4	1.2	96.4
		60.3	1.6	
		48.2	2.0	
		40.2	2.4	
		34.5	2.8	
Range of Motion	10.3	41.5	1.2	49.8
	13.8		1.6	66.4
	17.2		2.0	83.0
	20.7		2.4	99.6
	24.1		2.8	116.2

**Table 3 sensors-23-00904-t003:** In Vitro Test Parameters.

Test Variable	Range of Motion (±°)	Average Angular Speed (°/s)	Cycle Period (s)	Angle Swept per Cycle (°)
Average Angular Speed	20.0	68.9	1.4	96.4
		53.6	1.8	
		43.8	2.2	
		37.1	2.6	
		32.2	3.0	
Range of Motion	6.9	41.5	0.8	32.2
	10.3		1.2	49.8
	13.8		1.6	66.4
	17.2		2.0	83.0
	20.7		2.4	99.6

## Data Availability

All data produced during this study and the associated MATLAB scripts used to process the data can be found in the University of Leeds Data Repository (https://doi.org/10.5518/1232).

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
