# Peer review of "Inertial Tracking System for Monitoring Dual Mobility Hip Implants In Vitro"

_sensors, 2023, doi:10.3390/s23020904_

Round 1
Reviewer 1 Report
Review
Inertial tracking system for monitoring dual-mobility hip implants in-vitro
The authors present a useful and interesting study that aimed the in vitro measurement of motions of a dual-mobility hip implant system. Authors report the development, calibration and validation of the experimental set-up and obtained excellent results. The study was well designed and conducted and materials and methods used were suitable for the purpose. The Results section is interesting and detailing of the performance of the set-up is equally interesting. Even though, contrarily to what is said by the authors, it is not scientifically correct to say that this study provides for the first time a method of determining the function and potential of failure mechanisms (should correct this statement). But, in fact, I believe the technology developed and validated can be used for such purposes.
Minor errors to correct:
- Last paragraph of page 7 (Reference Source nor Found);
- Last paragraph of page 11 (Reference Source nor Found).
The authors present results and the discussion of those in Section 3 of the paper. Therefore, Section 4 should be named as a Conclusion section, since no discussion is presented in this section as it is.
Undoubtedly, this paper must be published.
Reviewer 2 Report
The authors address an important subject related to the development, calibration and validation of an inertial tracking system that could be used to measure the kinematic parameters of a dual mobility implant. The paper content and presented results are relevant with positive impact in the field and recommend the paper for publications after some improvements and clarifications:
1.The proposed system aims to evaluate the relative movement between the pelvis and femoral bone using a sensory system based on an IMU sensor, in the text there is no reference related the influence of the person body movement (rotations) in the sensor measurements and how the developed algorithm can eliminate these inputs resulting in this way only the angles amplitudes that are of interest
- 2.The obtained results (precision) should be compared with the results obtained using other measurement solutions proposed and presented in the literature, also advantages and disadvantages related to other sensory solutions should be emphasized
- 3.Clarify/detail the aim of the paper and the author contribution/contributions
Other changes/corrections/suggestions to the text
- Line 17 – in my opinion the “low-cost” mention in the abstract is less important than many other characteristics of the system, for example – dimensions, accuracy, repeatability, measurement principle etc.
- Line 41-43 – please motivate the necessity to measure these kinematic parameters of the dual mobility hip replacements from the medical point of view
- Line 72-77 – give bibliographic examples related to the paragraph statements
- Line 80-83 – the aim and contribution of the paper should be clearly presented
- The Introduction chapter should give a more details related to the state of the art
- Line 96-100 – give reference to the IMU datasheet file where the parameters are found
- Line 106-110 – text should be improved, it is not clear what exactly the C-shaped PCB contains (is the MCU on this PCB or is a separated component …)
- Line 114 – figure 2 is confusing, the diagram presents 3 different systems Tracker PCB + Controller + Host but in the image only the Tracker PCB is presented
- Line 142-143 – to general affirmation in the sentence related to “different calibration and test routines”, please mention which routines you applied
- Figure 5 – could be relevant to introduce the cartesian axis associated with the sensor and measured angles
- Line 212 – 211 – would be useful to quantify the error that the position variation of the sensor in the induced magnetic field is introducing, also mention which angles are affected by this variation
- Line 228 – please replace the text” Error! Reference source not found” with proper reference
- Line 243 – the correct text should be “in-vitro” not “in-vivo”
- Line 243 – 246 – please detail more the experiment parameters related with cycle period and angles amplitude
- Line 247-262 - please clarify the problem you want to solve here, is this related with the sensor or with the calibration procedure and the robot movement; the quaternion representation is provided by the sensor system or by the robot control system?
- Line 279 – 281 – Is not clear if the authors have identified this “manufacture induced errors”, what were the differences between the “five different boards” that you used
- Figure 7 – the range on peak angles on the plot could be reduced from -10…10 to -6 to 6
- Line 308 – link is not working, alternative way to share the results should be taken in account
- Figure 8 – would be nice to have the sensor system and solenoid presented in the image
- Line 339 – please replace the text” Error! Reference source not found” with proper reference
- Line 348 – the authors don’t use the journal correct way to introduce in text reference to an image
- Figure 9 – the range on peak angles on the plot could be reduced from -10…10 to -6 to 6
- Figure 10 – the range on peak angles on the plot could be reduced from -10…10 to -6 to 6
- Line 407 – link is not working, alternative way to share the results should be taken in account
- Line 409-411 – the experiments were conducted in laboratory conditions, “real-world conditions” could create confusion
- The Discussion chapter should address the also: the advantages and disadvantages of the proposed system; relevance of the obtained results in comparation with a real patient, results comparation with similar systems
Round 2
Reviewer 2 Report
The paper updates increased the quality of the paper and clarified the raised problems during first review round. I propose the paper for publication.